# Enzymatic Activity of Soil after Applying Mineral Fertilizers and Waste Lignite to Maize Grown for Silage

**Barbara Symanowicz \***[ID]**, Rafał Toczko** [ID] **and Martyna Toczko**

Institute of Agriculture and Horticulture, Faculty of Agrobioengineering and Animal Husbandry, Siedlce University of Natural Sciences and Humanities, B. Prusa 14, 08-110 Siedlce, Poland
\* Correspondence: barbara.symanowicz@uph.edu.pl

**Abstract:** This paper examines the effect of mineral fertilizers and waste lignite, the latter applied to the preceding crop, on the enzymatic activity of soil. The research was conducted between 2014 and 2016 at the Agricultural Experimental Station of the Siedlce University of Natural Sciences and Humanities in Zawady (Poland). The following treatment combinations were involved: 1—control; 2—NPKMgS; 3—NPKMgS + 20 N; 4—NPKMgS + 40 N; 5—NPKMgS + 60 N. Three varieties of maize grown for silage constituted the second research factor. Mineral fertilizers were applied at the following doses (pre-sowing): N-100, P-35, K-125, Mg-12 and S-14 kg ha$^{-1}$. Nitrogen was additionally applied as top dressing with 3, 4, and 5 treatment combinations at 20, 40, and 60 kg ha$^{-1}$. Waste lignite was applied to the preceding crop on two plots (3 and 4), in 1 and 5 t ha$^{-1}$ doses. In the subsequent years, the significantly highest content of soil organic carbon was recorded on the NPKMgS + 60 N plot (1.12%; 0.98% and 1.16%). With 49.25 and 51.95 mg NH$_4$-N h$^{-1}$ kg$^{-1}$ DM, urease activity in the soil treated with NPKMgS + 20 N and NPKMgS + 40 N was 56.95% and 65.55% higher than in the control plot soil. Throughout the experimental years, acid phosphatase activity did not vary much and amounted to 3.51–3.53 mmol PNP h$^{-1}$ kg$^{-1}$ DM. Mineral fertilizers significantly increased the activity of alkaline phosphatase and dehydrogenases in relation to the control. To ensure the high enzymatic activity of the soil and a high biochemical index of soil fertility, pre-sowing fertilizers at the level of 100 kg N, 35 kg P, 125 kg K, 12 kg Mg, and 14 kg S per hectare and top dressing of 20 kg N or 40 kg N per hectare are recommended. At the same time, it is advisable to use 1 t ha$^{-1}$ or 5 t ha$^{-1}$ of waste lignite of low energy value on the preceding crop.

**Keywords:** soil pH; soil organic carbon; enzyme activities; biochemical index of soil fertility; fertilization with NPKMgS; waste lignite; maize

## 1. Introduction

The intensification of agricultural production and increase of mineral fertilizer use contribute to the accelerated mineralization of organic matter and soil humus, which is the main causes of degradation of agricultural soils. Soil nutrients and organic matter are important factors influencing soil microbe composition [1–4]. Therefore, the use of waste lignite with a low energy value is an excellent way to supply organic carbon compounds to the soil. Waste organic lignite is characterized by high durability and resistance to decomposition and mineralization. It can be used on light sandy soils once every 8–10 years. In its composition, waste lignite contains (g kg$^{-1}$ DM): 680–710 organic carbon; 4–5 nitrogen; 0.01–0.14 phosphorus; 0.5–1.6 potassium; 0.5–1.6 magnesium; 8–30 calcium; 4–1.8 sulfur. The fertilizing effect of waste lignite also results from the content of such compounds as humic acids, fulvic acids, humins, and bitumens. Functional groups -COOH and -OH play an important role in the structural structure of these compounds.

The soil is a living, mobile system that contains free, extracellular enzymes and those enzymes that are found in the cells of microorganisms. Studies by other authors have shown that the addition of organic matter on the soil in the form of organic fertilizers can

significantly increase biological activity [5]. The amount and type of fertilization in the cultivation of maize also affects the increase in enzyme activity and the potential growth and development of plants [6].

Due to the high cost of mineral fertilizers, there is a growing interest in alternative solutions. Positive effects, including an increase in soil organic matter and major plant nutrients, can be obtained by using sewage sludge (SS) as fertilizer [7,8] and waste lignite [9,10]. Complete fertilization of maize grown for silage is poorly investigated and no studies on the impact of mineral and organic fertilizers on the enzymatic activity have been conducted, yet. Fertilizers modify the biological and chemical of the soil [1,11,12]. Soil bioactivity is a result of synergy between the activity of plants and the soil microorganisms [13]. The level of organic matter in the soil and fertilization determine the indicator of biological fertility and the activity of soil enzymes [3,5,6,11].

Urease is responsible for the biological hydrolysis of urea to $NH_3$ and $CO_2$, and an increase in soil pH. According to Yang et al. [14] and Zhao et al. [15] the activity of urease depends mainly on the type of organic and mineral fertilization.

Acid and alkaline phosphatases and dehydrogenases are considered the best indicators of the general population of soil microbial activity [16]. Dehydrogenases enzyme activity is commonly used as an indicator of soil biological activity. The highest activities of dehydrogenases and acid and alkaline phosphatases were noted on a plot with high content of organic matter. Phosphatases are a broad group of enzymes that catalyze hydrolysis of esters and anhydrides of phosphoric acid [17]. According to Steinweg et al. [18] low soil moisture can greatly reduce the enzymatic activity in the soil. The highest enzymatic activity was found in the soil, which was characterized by a high content of organic carbon and total nitrogen [4,19,20]. According to Cui and Holden [21] and Olivera et al. [22], carbon-related enzymes exhibited increased activity only in the deeper soil layers at 10–20 cm. Enzyme activity was linked to soil organic carbon content, rainfall, and irrigation [3]. Bielińska et al. [19] observed higher urease, phosphatases, and dehydrogenases activity in soils during sheep grazing. Soil enzymatic activity is also an important indicator for assessing and protecting biodiversity in important natural Natura 2000 habitats.

However, the effect of interaction between mineral fertilizers currently used in Polish and world agriculture and organic matter from waste lignite on soil enzyme activities has not been studied yet. A research hypothesis suggests that mineral fertilizers and waste lignite applied to the preceding crop may increase enzymatic activity.

The objective of this study was to determine the impact of mineral fertilizer and waste lignite, applied to the preceding crop (maize for silage), on organic carbon content, the activity of selected soil enzymes, and biochemical index of soil fertility.

## 2. Materials and Methods

### 2.1. Study Area

The field experiments were conducted on *Albic Luvisol (Arenic)* according to the WRB World reference of soil resources [23]. The research was conducted between 2014–2016 at the Agricultural Experimental Station of the Siedlce University of Natural Sciences and Humanities in Zawady—52°03′ N 22°33′ (Figure 1). The first factor was five fertilizer treatment combinations: (control (1), NPKMgS (2), NPKMgS + N1 (3), NPKMgS + N2 (4), NPKMgS +N3) (5). The second factor was three cultivars of maize: early, medium early, and medium late. Mineral fertilizers were applied before sowing maize at the following doses: 100 N, 35 P, 125 K, 12 Mg, and 14 S kg ha$^{-1}$ in the form of polyfoska® M-MAKS (NPKMgS), potassium salt 60% $K_2O$, and urea 46% N. Nitrogen was applied as top dressing with 3, 4, and 5 fertilizer combinations at doses of 20, 40, and 60 kg N ha$^{-1}$ in the form of urea 46% N. The chemical compounds of polyfoska® M-MAKS (NPKMgS): $NH_4H_2PO_4$, $(NH_4)_2HPO_4$, KCl, $MgCO_3$ and $K_2SO_4$. Waste lignite was applied to the preceding crop (maize grown for silage) in two doses (1 and 5 t ha$^{-1}$). The plant used in the three consecutive years of research was maize grown for silage. The experiment was conducted using a split-plot block design with two research factors, in three replications. All plot sizes were 15 m$^2$.

There was a distance of 2 m between the plots. Soil pH was 5.52–5.68 [10]. Total carbon and nitrogen content in the soil was determined with a Perkin Elmer (Waltham, MA, USA) CHNS/O 2400 auto analyzer coupled with a thermal conductivity detector (TCD), and by using acetanilide as the reference material. Total carbon was 10.53–13.68 g kg$^{-1}$ and total nitrogen was 1.08–1.32 g kg$^{-1}$, respectively [10]. The content of available phosphorus and potassium in the soil, determined with the Egner–Riehm method DL, and magnesium, determined with the Schatschabel method, was 55–78 mg kg-1, 123–168 mg kg$^{-1}$ and 61–74 mg kg$^{-1}$, respectively [10]. The content of available forms of P, K, Mg in the soil was determined with the ICP-AES method, with an inductively excited plasma atomic emission spectrometer (optima 3200RL, Perkin Elmer, Waltham, MA, USA).

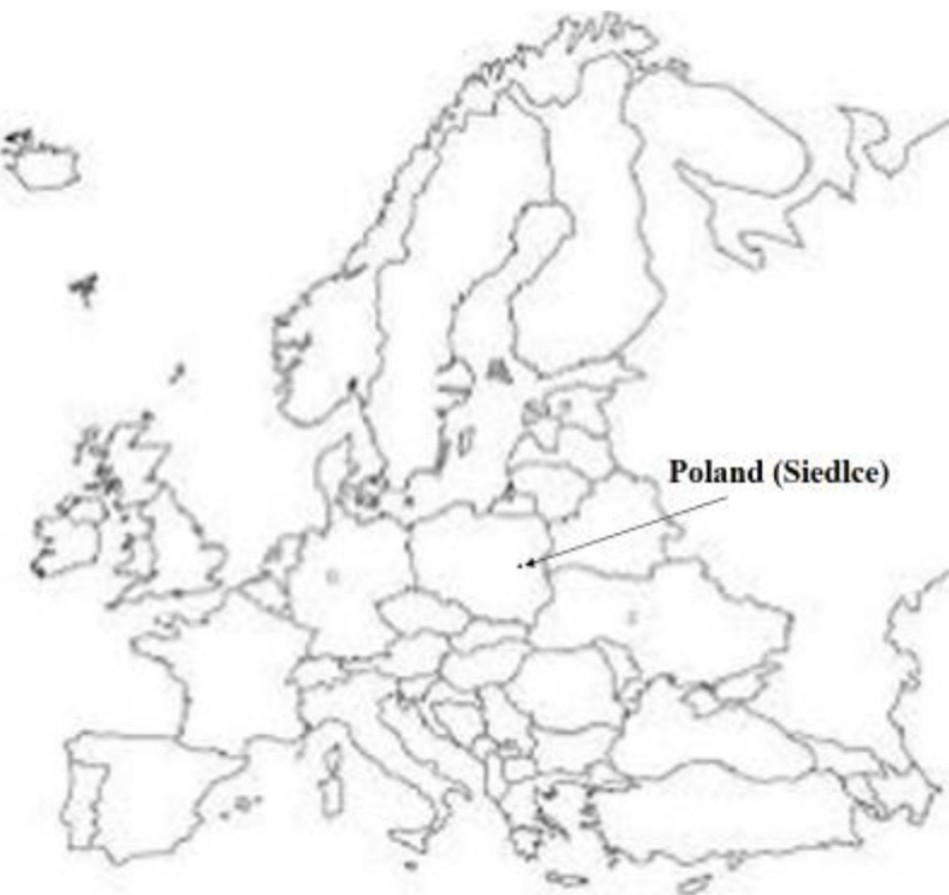

**Figure 1.** Location of the research site (https://contourmaps.com.pl (accessed on 17 February 2019) as modified by the authors.

*2.2. Soil Sampling and Analyses*

Soil samples were collected from a depth of 0–30 cm, after maize was harvested. Organic carbon (OC) was determined by applying the oxidation-titrimetric method according to Kalembasa [24]. In this method, soil, potassium dichromate, and a mixture of acids (sulfuric and phosphoric acid—5:1) are added. When the color changes to green, assuming the green color has been corrected, the indicator (N-phenylanthranilic acid) is added and the suspension in the flask is titrated with Mohr's salt. The pH of soil was determined on the basis of the basic potentiometric measurement in 1M KCl, with a soil solution ratio of 1:2.5 (*m/v*). The content of mineral forms of nitrogen (N-NH$_4^+$ and N-NO$_3^-$) was determined by flow colorimetry [25].

Urease (URE) (EC 3.5.1.5) activity was determined by a colorimetric method, according to Alef and Nannipieri [26]. The soil was incubated with an aqueous urea solution. After incubation, citrate buffer was added, and urease activity was determined.

The activity of acid phosphatase (ACP) (EC 3.1.3.2) and alkaline phosphatase (ALP) (EC 3.1.3.1) was determined with a method using soil and a substrate. The Page [27] method with disodium 4-nitrophenol phosphate hexahydrate in a modified universal buffer (MUB) was used to determine acid phosphatase activity (at pH 6.5) and alkaline phosphatase activity (at pH 11). The intensity of the color (yellow) caused by the released p-nitrophenol was then measured spectrophotometrically.

The dehydrogenases (DHA) (EC 1.1.1) activity was determined by a colorimetric method according to Casida et al. [28]. During the incubation, TTC (2,3,5-triphenol tetrazolium chloride) was used as substrate reduced to TPF (triphenylformazan). All compounds were assayed colometrically on a Lambda 25 UV-VIS spectrophotometer (Perkin Elmer, Waltham, MA, USA).

*2.3. Biochemical Index of soil Fertility (BI)*

The (BI) was calculated according to Kucharski et al. [29]:

BI = organic carbon content (%) (urease $\times 10^{-1}$ + acid phosphatase + alkaline phosphatase + dehydrogenases)

*2.4. Weather Conditions*

Characteristics of hydro-meteorological conditions were provided by the Institute of Meteorology and Water Management in Warsaw, the Hydrological and Meteorological Station in Siedlce (Figure 2). During the period of investigation, the mean temperatures of air was not differentiated in 2014, 2015, and 2016 years. The amount of rainfall varied across the years of research, with the highest in 2016 and the lowest in 2014 years. Weather conditions significantly affected the enzymatic activity of soil. The highest average temperatures were in July and August. The lowest amount of rainfall was recorded in 2014 years (mean 402 mm).

*2.5. Statistical Analysis*

The results obtained were subjected to statistical analysis using the analysis of variance ANOVA (Statistica 13.1 StatSoft, Inc., Tulsa, OK, USA). The least significant difference (LSD) determined by the Tukey's test. The criterion for significance was set at $p \leq 0.05$. The correlation coefficients between selected features were determined (Statistica 13.1 StatSoft, Inc., Tulsa, OK, USA).

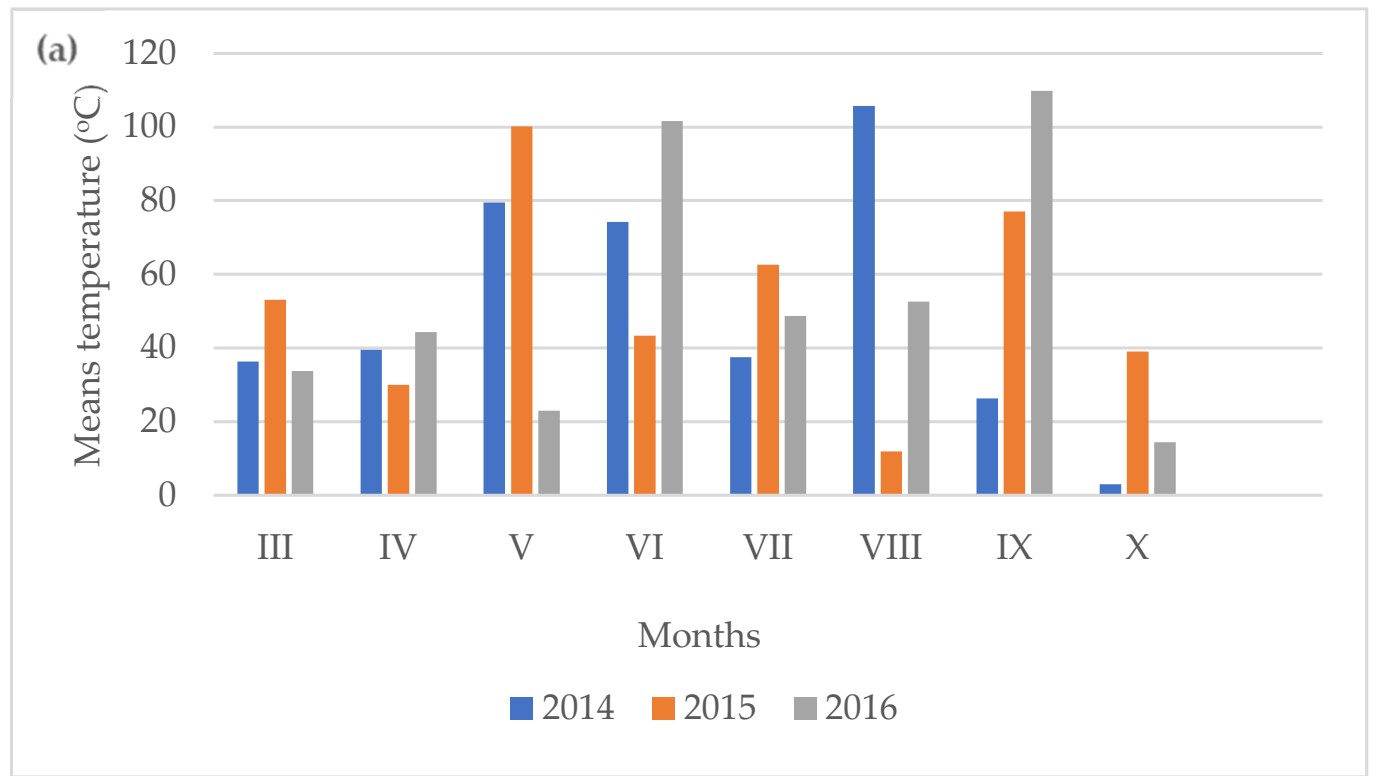

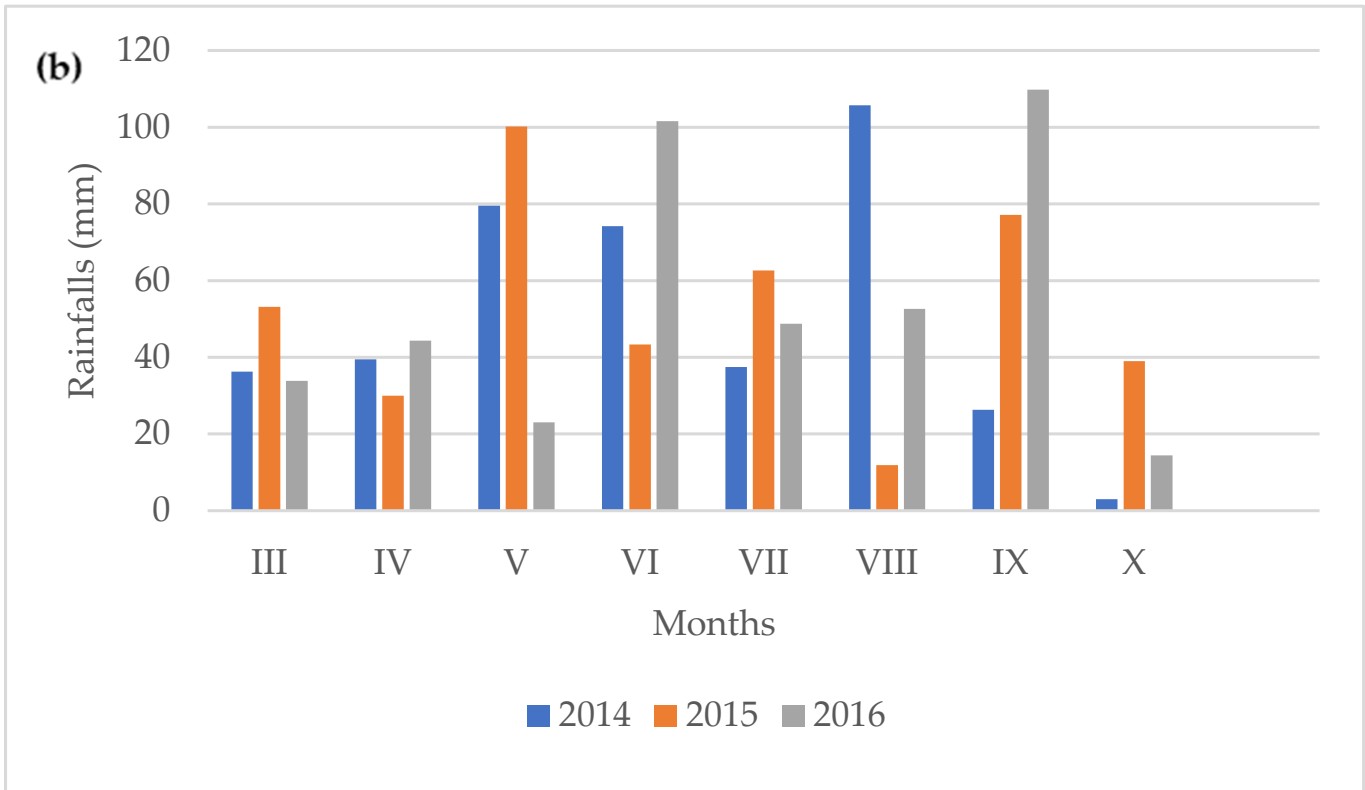

**Figure 2.** The air temperature (**a**) and rainfall (**b**) during vegetation period noted by Hydrological and Meteorological Station in Siedlce (Poland).

## 3. Results

### 3.1. Soil pH

The three-year experiment was conducted on soil with granulometric composition of loamy sand. Soil sampled from the control plot and from that treated with NPKMgS was very acidic (pH in mol dm$^{-3}$ KCl of 4.95–5.37). On plots treated with NPKMgS + 20 N, NPKMgS + 40 N and NPKMgS + 60 N (Table 1) slightly acidic soil reaction was recorded. On those plots, respectively, 1 t ha$^{-1}$ and 5 t ha$^{-1}$ of waste lignite and 30 t ha$^{-1}$ of manure were applied to the preceding crop.

**Table 1.** Soil pH in the years of research.

| Treatment | Research Years | | |
|---|---|---|---|
| | **2014** | **2015** | **2016** |
| Control [1] | 4.95—very acidic | 5.37—very acidic | 5.23—very acidic |
| NPKMgS | 5.30—very acidic | 5.52—very acidic | 5.48—very acidic |
| NPKMgS + 20 N | 5.67—slightly acidic | 6.38—slightly acidic | 6.11—slightly acidic |
| NPKMgS + 40 N | 5.67—slightly acidic | 6.42—slightly acidic | 6.26—slightly acidic |
| NPKMgS + 60N | 6.38—slightly acidic | 6.50—slightly acidic | 6.01—slightly acidic |

[1] NPKMgS–100 N, 35 P, 125 K, 12 Mg, 14 S kg ha$^{-1}$.

### 3.2. Mineral Forms of Nitrogen

The content of mineral nitrogen, i.e., of ammonium and nitrate forms, in the soil was determined in the spring in subsequent years of research and presented as three-years means (Table 2). The two soil layers (0–30 cm and 30–60 cm) indicated a very low level of nitrogen forms on the control plot and very high on those treated with NPKMgS + 40 N and NPKMgS + 60 N. On NPKMgS, NPKMgS + 20 N, and the control plots, higher content of ammonium and nitrate forms in the humus layer (0–30 cm) was recorded than in the lower layer. However, in the soil from NPKMgS + 40 N and NPKMgS + 60 N plots, higher amounts of plant-available nitrogen forms were recorded in the layer of 30–60 cm. Their amounts in the soil was the largest on the NPKMgS + 40 N plot and amounted to 114.16 kg ha$^{-1}$, nearly four times as much as on the control plot (33.54 kg ha$^{-1}$).

**Table 2.** Mineral nitrogen in soil (3-year average).

| Treatment | Depth cm | $N\text{-}NO_3^-$ mg kg$^{-1}$ | $N\text{-}NH_4^+$ mg kg$^{-1}$ | $N\text{-}NO_3^-$ and $N\text{-}NH_4^+$ | Sum mg kg$^{-1}$ | Conversion Factor [2] | $N_{min}$ kg ha$^{-1}$ | Abudance |
|---|---|---|---|---|---|---|---|---|
| Control [1] | 0–30 | 3.02 | 1.83 | 4.85 | 7.80 | 4.30 | 33.54 | very low |
| | 30–60 | 0.84 | 2.11 | 2.95 | | | | |
| NPKMgS | 0–30 | 3.81 | 2.62 | 6.43 | 9.65 | 4.30 | 41.49 | low |
| | 30–60 | 0.53 | 2.69 | 3.22 | | | | |
| NPKMgS + 20 N | 0–30 | 7.38 | 3.42 | 10.80 | 17.33 | 4.30 | 74.52 | medium |
| | 30–60 | 1.22 | 5.31 | 6.53 | | | | |
| NPKMgS + 40 N | 0–30 | 9.85 | 2.93 | 12.78 | 26.55 | 4.30 | 114.16 | very high |
| | 30–60 | 11.24 | 2.53 | 13.77 | | | | |
| NPKMgS + 60 N | 0–30 | 8.49 | 3.11 | 12.60 | 25.48 | 4.30 | 109.56 | very high |
| | 30–60 | 11.31 | 2.57 | 13.88 | | | | |

[1] NPKMgS–100 N, 35 P, 125 K, 12 Mg, 14 S kg ha$^{-1}$; [2] for medium soils [12].

### 3.3. Organic Carbon

The average organic carbon concentration of the soil was 0.92% (Table 3). In a statistically significant way, the most organic carbon was recorded in the soil from NPKMgS + 20 N, NPKMgS + 40 N plots, to which, respectively, 1 t ha$^{-1}$ and 5 t ha$^{-1}$ of waste lignite were applied to the preceding crop (maize grown for silage). Those amounts of waste lignite applied to the soil increased its organic carbon concentration by 23.7 and 38.7% compared to the control. The high content of organic carbon in the soil taken from the plots where waste lignite was used for the preceding crop indicates a high carbon content in this waste. Maize varieties did not significantly affect soil organic carbon content. Statistical analysis showed a significant variation of soil organic carbon content between the first and second year and the first and third year. Across the three experimental years, the significantly highest content of soil organic carbon was recorded on the NPKMgS + 40 N plot (1.12%; 0.98% and 1.16%). On this plot, 5 t ha$^{-1}$ of waste lignite were applied to the preceding crop. Significantly, the most organic carbon in the soil was recorded in the first year, but in the following years of research the content slightly decreased.

**Table 3.** Total organic carbon in soil (%).

| Treatment | Cultivars | | | Research Years | | |
|---|---|---|---|---|---|---|
| | Early | Medium Early | Medium Late | 2014 | 2015 | 2016 |
| Control [1] | 0.83 Ba | 0.89 Aa | 0.81 Ba | 0.84 Ba | 0.72 Ba | 0.69 Ca |
| NPKMgS | 0.83 Ba | 0.81 Ba | 0.87 Aa | 0.84 Ba | 0.84 Aa | 0.80 Ba |
| NPKMgS + 20 N | 1.03 Aa | 1.03 Aa | 0.96 Aa | 1.01 Aa | 0.94 Aa | 0.95 Ba |
| NPKMgS + 40 N | 1.20 Aa | 1.11 Aa | 1.07 Aa | 1.12 Aa | 0.98 Aa | 1.16 Aa |
| NPKMgS + 60 N | 0.96 Aa | 0.96 Aa | 0.88 Aa | 0.93 Aa | 0.80 Ba | 0.87 Ba |

[1] NPKMgS–100 N, 35 P, 125 K, 12 Mg, 14 S kg ha$^{-1}$. Different uppercase letters within a column for fertilizer and different lowercase letters within a line indicate significant differences ($p \leq 0.05$).

### 3.4. Urease Activity

Organic matter introduced into the soil in the form of waste lignite and manure had a decisive effect on the level of urease activity. Fertilizer treatment, varieties of maize grown for silage and years of research significantly differentiated the activity of urease in the soil. Its mean value was 45.02 mg NH$_4$-N h$^{-1}$ kg$^{-1}$ DM (Table 4).

**Table 4.** Urease activity in soil (mg NH$_4$-N h$^{-1}$ kg$^{-1}$ DM).

| Treatment | Cultivars | | | Research Years | | |
|---|---|---|---|---|---|---|
| | Early | Medium Early | Medium Late | 2014 | 2015 | 2016 |
| Control [1] | 32.38 Da | 31.52 Eb | 30.07 Dc | 30.94 Ea | 31.42 Ea | 31.95 Ea |
| NPKMgS | 37.45 Ca | 36.95 Da | 37.23 Ca | 37.91 Db | 36.57 Dc | 38.92 Da |
| NPKMgS + 20 N | 49.17 Bb | 49.05 Cb | 50.05 Ba | 48.16 Cb | 49.1 Ca | 49.93 Ca |
| NPKMgS + 40 N | 51.25 Ba | 52.01 Ba | 52.41 Ba | 52.11 Ba | 51.53 Ba | 52.42 Ba |
| NPKMgS + 60 N | 54.73 Ab | 54.57 Ab | 55.64 Aa | 55.45 Aa | 54.63 Aa | 55.21 Aa |

[1] NPKMgS–100 N, 35 P, 125 K, 12 Mg, 14 S kg ha$^{-1}$. Different uppercase letters within a column for fertilizer and different lowercase letters within a line indicate significant differences ($p \leq 0.05$).

In a statistically significant way, the greatest activity of urease was recorded in the soil treated with NPKMgS + 60 N. On that plot, the preceding crop was additionally treated with manure at a dose of 30 t ha$^{-1}$. With 49.25 and 51.95 mg NH$_4$-N h$^{-1}$ kg$^{-1}$ DM, urease activity in the soil treated with NPKMgS + 20 N and NPKMgS + 40 N was 56.95% and 65.55% higher than in the control plot soil. With the mean value of 44.97 mg NH$_4$-N h$^{-1}$ kg$^{-1}$ DM, urease activity across maize varieties did not significantly vary. Yet, significant differences were noted for the interaction between a variety and fertilizer plots. Significantly, urease highest activity was on the plot with the medium late variety treated with NPKMgS + 60 N. Throughout the experiment, the value was significantly higher in the third year than in the first and second.

### 3.5. Acid Phosphatase Activity

Table 5 presents the effects of fertilizer treatment and varieties on acid phosphatase activity across the experimental years. Its mean value across those factors was 3.52 mmol PNP h$^{-1}$ kg$^{-1}$ DM. Mineral fertilizer treatment significantly reduced the activity of acid phosphatase. The largest reduction (by 16% in relation to control) was recorded after the application NPKMgS + 60 N. This fact should be combined with a reduction in soil acidity in this plots. The highest activity of acid phosphatase was in the soil on which the early variety was grown. Throughout the experimental years, its activity was at a similar level and amounted to 3.51–3.53 mmol PNP h$^{-1}$ kg$^{-1}$ DM. The highest level of acid phosphatase activity was in the soil from the control plots.

**Table 5.** Acid phosphatase activity in soil (mmol PNP h$^{-1}$ kg$^{-1}$DM).

| Treatment | Cultivars | | | Research Years | | |
|---|---|---|---|---|---|---|
| | Early | Medium Early | Medium Late | 2014 | 2015 | 2016 |
| Control [1] | 4.09 [Aa] | 3.84 [Aa] | 3.74 [Aa] | 3.96 [Aa] | 3.81 [Aa] | 3.84 [Aa] |
| NPKMgS | 3.85 [Aa] | 3.61 [Aa] | 3.51 [Aa] | 3.67 [Aa] | 3.53 [Ba] | 3.63 [Aa] |
| NPKMgS + 20 N | 3.47 [Ba] | 3.53 [Aa] | 3.38 [Ba] | 3.31 [Ba] | 3.47 [Ba] | 3.45 [Ba] |
| NPKMgS + 40 N | 3.52 [Ba] | 3.28 [Ba] | 3.45 [Ba] | 3.37 [Ba] | 3.45 [Ba] | 3.41 [Ba] |
| NPKMgS + 60 N | 3.26 [Ba] | 3.13 [Ba] | 3.26 [Ba] | 3.22 [Ba] | 3.38 [Ba] | 3.32 [Ba] |

[1] NPKMgS–100 N, 35 P, 125 K, 12 Mg, 14 S kg ha$^{-1}$. Different uppercase letters within a column for fertilizer and different lowercase letters within a line indicate significant differences ($p \leq 0.05$).

### 3.6. Alkaline Phosphatase Activity

The activity of alkaline phosphatase must be combined with the acidity of the soil. Of the experimental factors, only fertilizer treatment significantly differentiated the activity of alkaline phosphatase. Its mean value was 0.71 mmol PNP h$^{-1}$ kg$^{-1}$ DM (Table 6). Mineral fertilizers significantly increased the activity of alkaline phosphatase in relation to control. Its highest value, with an increase of 22.58% compared to control, was recorded in the soil from plots treated with NPKMgS + 40 N and NPKMgS + 60N. On these plots, 5 t ha$^{-1}$ of waste lignite and 30 t ha$^{-1}$ of manure were applied for the preceding crop, respectively. The pH of the soil on these plots was also lower.

**Table 6.** Alkaline phosphatase activity in soil (mmol PNP h$^{-1}$ kg$^{-1}$DM).

| Treatment | Cultivars | | | Research Years | | |
|---|---|---|---|---|---|---|
| | Early | Medium Early | Medium Late | 2014 | 2015 | 2016 |
| Control [1] | 0.64 Ca | 0.59 Ca | 0.63 Da | 0.61 Ca | 0.63 Ca | 0.62 Ca |
| NPKMgS | 0.71 Ba | 0.69 Ba | 0.67 Ca | 0.69 Ba | 0.71 Ba | 0.70 Ba |
| NPKMgS + 20 N | 0.65 Ca | 0.72 Ba | 0.69 Ca | 0.71 Bb | 0.75 Aa | 0.72 Bb |
| NPKMgS + 40 N | 0.78 Aa | 0.76 Aa | 0.73 Bb | 0.76 Aa | 0.78 Aa | 0.77 Aa |
| NPKMgS + 60 N | 0.73 Bc | 0.75 Ab | 0.79 Aa | 0.76 Aa | 0.77 Aa | 0.75 Aa |

[1] NPKMgS–100 N, 35 P, 125 K, 12 Mg, 14 S kg ha$^{-1}$. Different uppercase letters within a column for fertilizer and different lowercase letters within a line indicate significant differences ($p \leq 0.05$).

### 3.7. Dehydrogenases Activity

The activity of dehydrogenases in the soil was significantly differentiated as a response to the experimental factors (Table 7). Significantly, the highest activity was found in the soil treated with NPKMgS + 40 N. On this plot, 5 t ha$^{-1}$ of waste lignite was applied to the preceding crop. Mineral fertilizers significantly increased the activity of dehydrogenases. The greatest value was observed in the soil on which the medium late variety was grown. In the first year of the experiment, the activity of dehydrogenases was higher than in the second and third. Statistical analysis showed the highest activity of dehydrogenases in the soil fertilized with NPKMgS + 40 N, on which the medium late variety was grown.

**Table 7.** Dehydrogenases activity in soil (cm$^3$ H$_2$ h$^{-1}$ kg$^{-1}$ DM).

| Treatment | Cultivars | | | Research Years | | |
|---|---|---|---|---|---|---|
| | Early | Medium Early | Medium Late | 2014 | 2015 | 2016 |
| Control [1] | 31.6 Da | 32.1 Ca | 32.8 Ca | 32.4 Ca | 33.7 Ca | 33.4 Ca |
| NPKMgS | 32.4 Db | 34.5 Ca | 35.1 Ca | 34.3 Ca | 34.1 Ca | 34.2 Ca |
| NPKMgS + 20 N | 38.5 Ca | 39.3 Ba | 40.3 Ba | 39.1 Ba | 38.9 Ba | 39.0 Ba |
| NPKMgS + 40 N | 45.2 Aa | 44.8 Aa | 45.4 Aa | 45.7 Aa | 43.6 Aa | 44.1 Aa |
| NPKMgS + 60 N | 41.1 Bb | 43.4 Aa | 44.2 Aa | 43.9 Aa | 43.0 Aa | 43.4 Aa |

[1] NPKMgS–100 N, 35 P, 125 K, 12 Mg, 14 S kg ha$^{-1}$. Different uppercase letters within a column for fertilizer and different lowercase letters within a line indicate significant differences ($p \leq 0.05$).

### 3.8. Biochemical Index of Soil Fertility (BI)

Table 8 presents the effect of fertilizer treatment and varieties on the (BI) across experimental years. Mineral fertilizer treatment with NPKMgS + 40 N significantly affected the index value. On that plot, to which 5 t ha$^{-1}$ of waste was applied to the preceding crop, it reached 60.15 and was 87.55% higher than the control plot value. Significantly, the highest average value of (BI) was recorded for the soil on which the medium early variety was grown. It was the highest (44.90) in the first year of research.

**Table 8.** Biochemical index of soil fertility.

| Treatment | Cultivars | | | Research Years | | |
|---|---|---|---|---|---|---|
| | Early | Medium Early | Medium Late | 2014 | 2015 | 2016 |
| Control [1] | 32.84 Db | 35.32 Ca | 32.54 Db | 33.65 Da | 29.72 Eb | 28.33 Db |
| NPKMgS | 33.78 Db | 34.42 Cb | 37.41 Ca | 35.14 Da | 35.28 Da | 33.94 Ca |
| NPKMgS + 20 N | 39.53 Cc | 49.91 Ba | 47.40 Bb | 40.11 Cb | 45.15 Ba | 45.75 Ba |
| NPKMgS + 40 N | 65.55 Aa | 59.98 Ab | 58.66 Ab | 62.20 Aa | 52.45 Ab | 62.08 Aa |
| NPKMgS + 60 N | 48.54 Bb | 50.63 Ba | 47.36 Bb | 53.42 Ba | 42.67 Cc | 46.10 Bb |

[1] NPKMgS–100 N, 35 P, 125 K, 12 Mg, 14 S kg ha$^{-1}$. Different uppercase letters within a column for fertilizer and different lowercase letters within a line indicate significant differences ($p \leq 0.05$).

### 3.9. Linear Correlation between OC, Soil Enzyme Actvities and BI

On the basis of the research, significant relationships were found between the activity of the enzymes at $p \leq 0.05$ and $p \leq 0.01$ (Table 9). The ACP was significantly negatively correlated with the OC, URE, ALP, DHA, and BI. The positive correlations were noted between the URE and the OC, and between the ALP and OC and URE. Significant relationships occurred between the DHA and OC, URE, and ALP. The BI was significantly positively correlated with the OC, URE, ALP, and DHA.

**Table 9.** Linear correlation coefficients between OC, URE, ACP, ALP, DHA, and BI.

| Variable | OC | URE | ACP | ALP | DHA | BI |
|---|---|---|---|---|---|---|
| OC | 1.00 | | | | | |
| URE | 0.73 * | 1.00 | | | | |
| ACP | −0.62 * | −0.98 ** | 1.00 | | | |
| ALP | 0.72 * | 0.95 ** | −0.95 ** | 1.00 | | |
| DHA | 0.83 * | 0.95 ** | −0.89 ** | 0.93 ** | 1.00 | |
| BI | 0.93 ** | 0.85 * | −0.76 * | 0.87 * | 0.96 ** | 1.00 |

* significant for $p \leq 0.01$; ** significant for $p \leq 0.05$.

## 4. Discussion

Enzymatic activity of the soil depends to the greatest extent on its content of organic matter and nitrogen, its pH and temperature, on mineral and organic fertilizer treatment, on the content of macronutrient, micronutrient, and heavy metals, but also on fertilizer treatment applied to the preceding crop [30–35]. Enzymatic activity in soil is related to its high content of mineral nitrogen (Table 2) and organic carbon (Table 3). The present research confirmed a significant effect of organic carbon introduced into the soil with waste lignite and manure on an increase in urease activity. This finding has been confirmed by other studies [36–39]. The urease activity determined in our own research was weak or very weak. This proves the low content of mineral nitrogen in the soil and the low pH. According to Maphuhla et al. [34] low urease activity may also be related to slower decomposition of organic matter. Then the release of nitrogen is less compared to the amount of nitrogen absorbed by the microorganisms. The optimal application of mineral fertilizers and organic matter in the form of waste lignite to maize grown for silage in the present research had a positive effect on the level of soil enzymatic activity. The results confirmed the research of other authors [19–21]. Moreover, the positive effect of mineral fertilizers and organic waste materials on the stimulation of enzymatic activity was presented by Tan et al. [17] and Medina et al. [40].

Additionally, according to Możdżer [41], an increase in the enzymatic activity of the soil can be achieved after the use of mineral fertilizers and ash granules produced after the combustion of sewage sludge. In those studies, the stimulating effect of fertilizers, in the range of 7.41–30.5%, on the activity of urease, phosphatases, and dehydrogenases was demonstrated. Containing large amounts of organic matter, sewage sludge used in high doses (up to 60 t DM ha$^{-1}$) significantly increased the activity of urease, acid phosphatase and dehydrogenases compared to the activity of those enzymes recorded in the control plot soil [42]. In the present research, the source of organic matter was waste lignite and manure.

Acid and alkaline phosphatase are enzymes that are very often used to assess the ecological status of soils [43]. The activity of phosphatases is also used to assess the amount of mineral forms of phosphorus needed to ensure a high level of soil fertility [43–45].

In the present studies, it turned out that reduced values of soil chemical parameters, like pH and organic carbon content, were correlated with lower activity of alkaline phosphatase and dehydrogenases. The acid reaction of the soil in the control plots and plots fertilized with NPKMgS resulted in an increase in acid phosphatase activity in the subsequent years of research. The enzymatic activity of soil usually increases with increasing soil pH [46–48]. According to Acosta-Martinez and Tabatabai [49], soil pH 5.6–5.8 was optimal for the acid phosphatase and dehydrogenases activity. Studies by many authors [17,40] have shown that the application of municipal sewage sludge (SS) to soil increases enzymatic activity. The increase in enzyme activity is associated with a large amount of plant nutrients, organic matter and a significant amount of microorganisms introduced into the soil from SS [40]. The slightly acid reaction of the soil from the plots fertilized with NPKMgS, NPKMgS + 20 N, NPKMgS + 40 N, and NPKMgS + 60 N increased the activity of alkaline phosphatase in relation to the soil from the control plot. Zieleniewicz et al. [50] indicated that a decrease in alkaline phosphatase and dehydrogenases activity might have been due to the use of calcium sulfate at a dose of 500 kg ha$^{-1}$. The decrease in the activity of these enzymes was a direct result of a decrease in soil pH. In other studies by the same authors [51], with grass mixtures and legume–grass mixtures, it was found that NPK fertilizers and the Physioactiv biostimulant had a positive effect on the activity of dehydrogenases and alkaline phosphatase

The present experiment with maize grown for silage confirmed a significant effect of mineral fertilizers and organic matter on soil enzymatic activity [52]. An increase in enzyme activity as a response to organic matter application to the soil was also reported by Widmer et al. [53] and Xie et al. [54].

A significant correlation between the enzymatic activity in soils was reported by other authors [31,34,43]. Correlations between soil enzyme content and pH and carbon content was observed by other authors [19,43,52].

## 5. Conclusions

The mineral fertilizers applied to maize grown for silage and waste lignite applied to the preceding crop increased the activities of urease, phosphatases, and dehydrogenases. In order to obtain high enzymatic activity of the soil and a high biochemical index of soil fertility, pre-sowing fertilizers at the level of 100 kg N, 35 kg P, 125 kg K, 12 kg Mg, and 14 kg S per hectare and top dressing of 20 kg N or 40 kg N per hectare are recommended. At the same time, it is advisable to use 1 t ha$^{-1}$ or 5 t ha$^{-1}$ of waste lignite of low energy value to the preceding crop.

**Author Contributions:** Conceptualization, B.S. and M.T.; methodology, B.S.; validation, B.S., M.T. and R.T.; formal analysis, B.S. and M.T.; investigation, B.S., M.T. and R.T.; data curation, B.S., M.T. and R.T.; writing—original draft preparation, B.S. and M.T.; writing—review and editing, B.S. and M.T.; visualization, B.S., M.T. and R.T. All authors have read and agreed to the published version of the manuscript.

**Funding:** This research was financed from the science grant by the Polish Ministry of Education and Science, research task number 36/20/B.

**Institutional Review Board Statement:** Not applicable.

**Informed Consent Statement:** Not applicable.

**Data Availability Statement:** The data presented in this study are available on request from the corresponding author.

**Conflicts of Interest:** The authors declare no conflict of interest.

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
