# Peer review of "Enzymatic Activity of Soil after Applying Mineral Fertilizers and Waste Lignite to Maize Grown for Silage"

_agriculture, doi:10.3390/agriculture12122146_

Round 1

Reviewer 1 Report

After carefully reading the manuscript, I recommend this manuscript for publication in Agriculture after major revision. The main issue of manuscript “Enzymatic Activity of Soil after Applying Mineral Fertilizers and Waste Lignite of Maize Grown for Silage” is one of scientific context for environment, however, the manuscript lacked in many points.

 Some major recommendations:

1-     I would also recommend having the manuscript edited thoroughly for English usage. Your English is good enough, however a number of grammatical errors still exist and make your meaning unclear in some places.

2-     The introduction does not do an adequate job explaining the innovation of your work enough and lack in explore the novelty and the objective.

3-     The materials and method section needs more clarification:

a-      Why the author add more nitrogen 20N. 40N, and 60N

b-     The author mentioned, fertilizers were added before sowing, he can did that with P and waste lignite While with N, and K fertilizer we added them during the season to reduce the lost.

c-      It not clear when the author added 1 and 5t / h.

4-     The changing in soil organic carbon need to more clarification.

Author Response

After carefully reading the manuscript, I recommend this manuscript for publication in Agriculture after major revision. The main issue of manuscript “Enzymatic Activity of Soil after Applying Mineral Fertilizers and Waste Lignite of Maize Grown for Silage” is one of scientific context for environment, however, the manuscript lacked in many points.

Thank you for your review.

After carefully reading the comments and suggestions of the Reviewer kindly inform that the manuscript has been revised taking into account the legitimate Reviewer’s comments.

  • I would also recommend having the manuscript edited thoroughly for English usage. Your English is good enough, however a number of grammatical errors still exist and make your meaning unclear in some places.

BS response: Thank you very much for this attention. The English language has been corrected and checked by a native speaker.

  • The introduction does not do an adequate job explaining the innovation of your work enough and lack in explore the novelty and the objective.

BS response: In the Introduction section, information on the desirability and innovativeness of using waste lignite with low energy value has been supplemented. (attached file)

3-     The materials and method section needs more clarification:

  • Why the author add more nitrogen 20N. 40N, and 60N

BS response: The maize grown for silage is a plant with high fertilization requirements due to high yields of biomass. Fertilization recommendations indicate the need to divide the total dose of nitrogen in order to increase the effectiveness of fertilization with this component.

  • The author mentioned, fertilizers were added before sowing, he can did that with P and waste lignite While with N, and K fertilizer we added them during the season to reduce the lost.

BS response: Mineral fertilizers was applied before sowing maize at the following doses: 100 N, 35 P, 125 K, 12 Mg and 14 S kg ha-1 in the form polyfoska® M-MAKS (NPKMgS), potassium salt 60% K2O and urea 46% N. Nitrogen was applied as top dressing with 3, 4, and 5 fertilizer combinations at a doses 20, 40, and 60 kg N ha – 1 in the form urea 46% N. The waste lignite was applied to the preceding crop (maize grown for silage) in two doses (1 and 5 t ha – 1). The Reviewer's suggestion to combine fertilization with phosphorus and waste lignite was not possible because a compound fertilizer (NPKMgS) was used, and waste lignite was applied to the preceding crop.

  • It not clear when the author added 1 and 5t / h.

BS response: The waste lignite was applied to the preceding crop (maize grown for silage) in two doses (1 and 5 t ha – 1). Maize is a crop that can be grown in monoculture.

4-     The changing in soil organic carbon need to more clarification.

BS response:  The subsection "Organic Carbon" has been improved and supplemented. (attached file)

Best regards

Authors

Reviewer 2 Report

agriculture-2010119

This study mainly explores the Enzymatic Activity of Soil after Applying Mineral Fertilizers and Waste Lignite of Maize Grown for Silage. The experimental design is reasonable and the measured indexes are rich. However, there was no in-depth analysis of experimental data. There are still major problems with the manuscripts, including writing, format and chart distribution, etc. Therefore, I suggest that this article be resubmitted after modified or resubmitted to other journals.

1. The abstract needs major revision. The abstract is unclear, confusing, and finally lacks conclusive sentences.

2. Line 46: “[7, 8}” replace with “[7, 8]”.

3. Line 47: Only need a space after the word of grown.

4. The introduction section needs major revision. Introduction section is logically confused lead to this article is difficult to read. No indication of the importance of current research, and also no clarification of existing scientific issues.

5. I have a question, the soil index data obtained is sampled after silage maize harvest? If so, why is the preceding crop mentioned? Applications of waste lignite specific time clearly, how to apply? What is the content of each element of waste coal? This is more conducive to readers to understand.

6. The plot is too small for silage maize, so how do you avoid the exchange of nutrients and water between the two plots

7. Line 97-98: If the ranges are 5.52-5.8, delete “respectively”. If use “respectively”, write down the soil pH values for three years respectively.

8. Line 126: Add a period after the sentence ends.

9. The results should be analyzed from the effects of mineral fertilizer, maize varieties, different years and their interactions on different indicators. Much key information in the results and analysis section is not shown.

10. Line 190: “)n that plot” replace with “In that plot”.

11. Line 190: In the manuscript, you have to make it clear what are the indicators.

12. Table 4 is placed before heading 3.5.

13. Chart form is too single, recommended figure s and tables reasonable allocation.

14. The manuscript lacks the experimental site map and the meteorological data of the experimental site (including temperature, rainfall, etc.)

15. The discussion section needs to be rewritten. The discussion does not explain in depth the reasons for the results of this study. Discussion is an analysis of the causes of some phenomena and further explanation of these causes. In addition, the discussion not hold the key content, the discussion section should highlight the main research content of this paper. Discussion section should around your research focus, rather than simply listing others ' references.

16. Conclusion section: “The mineral fertilizers applied to maize grown for silage and waste lignite applied to the preceding crop (silage maize) had a positive impact on the activity of selected soil enzymes. This description needs modification and is confusing to read.

Author Response

agriculture-2010119

This study mainly explores the Enzymatic Activity of Soil after Applying Mineral Fertilizers and Waste Lignite of Maize Grown for Silage. The experimental design is reasonable and the measured indexes are rich. However, there was no in-depth analysis of experimental data. There are still major problems with the manuscripts, including writing, format and chart distribution, etc. Therefore, I suggest that this article be resubmitted after modified or resubmitted to other journals.

Thank you for your review.

After carefully reading the comments and suggestions of the Reviewer kindly inform that the manuscript has been revised taking into account the legitimate Reviewer’s comments.

  1. The abstract needs major revision. The abstract is unclear, confusing, and finally lacks conclusive sentences.

BS response: Thank you very much for this attention. The abstract has been corrected as suggested by the Reviewer’s. (attached file)

  1. Line 46: “[7, 8}” replace with “[7, 8]”.

BS response: The technical error has been corrected. (attached file)

  1. Line 47: Only need a space after the word of grown.

BS response: The technical error has been corrected. (attached file)

  1. The introduction section needs major revision. Introduction section is logically confused lead to this article is difficult to read. No indication of the importance of current research, and also no clarification of existing scientific issues.

BS response: Thank you very much for this attention. The Introduction section has been supplemented and improved. (attached file)

  1. I have a question, the soil index data obtained is sampled after silage maize harvest?If so, why is the preceding crop mentioned?Applications of waste lignite specific time clearly, how to apply? What is the content of each element of waste coal? This is more conducive to readers to understand.

BS response: Soil samples were collected from the Ap level (0-30 cm) after maize harvest for silage. The preceding crop was mentioned because waste lignite was used for this crop. The effect of waste lignite in the soil is long-term (8-10 years). Therefore, we investigated the delayed effect of waste lignite on enzyme activity in the soil.

  1. The plot is too small for silage maize, so how do you avoid the exchange of nutrients and water between the two plots

BS response: The plot size of 15 m2 complies with the requirements in Poland. A distance of 2 m was maintained between the plots.

  1. Line 97-98: If the ranges are 5.52-5.8, delete “respectively”. If use “respectively”, write down the soil pH values for three years respectively.

BS response: „respectively” - has been removed.

  1. Line 126: Add a period after the sentence ends.

BS response: The technical error has been corrected. (attached file)

  1. The results should be analyzed from the effects of mineral fertilizer, maize varieties, different years and their interactions on different indicators. Much key information in the results and analysis section is not shown.

BS response: In the Results section, the description of the influence of the tested factors on the level of determined soil enzymes has been supplemented and improved. (attached file)

  1. Line 190: “)n that plot” replace with “In that plot”.

BS response: The technical error has been corrected. (attached file)

  1. Line 190: In the manuscript, you have to make it clear what are the indicators.

BS response: There are no standardized indicators. Many factors have a significant impact on the activity of urease in the soil (pH, organic matter content, soil abundance in available nutrients, temperature e.t.c.)

  1. Table 4 is placed before heading 3.5.

BS response: The technical error has been corrected. (attached file)

  1. Chart form is too single, recommended figure s and tables reasonable allocation.

BS response: According to the Reviewer's suggestion, figures are included.

  1. The manuscript lacks the experimental site map and the meteorological data of the experimental site (including temperature, rainfall, etc.)

BS response: The manuscript includes a map of the experimental site and meteorological data during the experiment. (attached file)

  1. The discussion section needs to be rewritten. The discussion does not explain in depth the reasons for the results of this study. Discussion is an analysis of the causes of some phenomena and further explanation of these causes. In addition, the discussion not hold the key content, the discussion section should highlight the main research content of this paper. Discussion section should around your research focus, rather than simply listing others ' references.

BS response: The Discussion section has been improved and supplemented.

  1. Conclusion section: “The mineral fertilizers applied to maize grown for silage and waste lignite applied to the preceding crop (silage maize) had a positive impact on the activity of selected soil enzymes”. This description needs modification and is confusing to read.

BS response: The Conclusions has been modified as suggested by the Reviewer’s. (attached file)

Best regards

Authors
